# MCP-Radar: A Multi-Dimensional Benchmark for Evaluating Tool Use Capabilities in Large Language Models

## Abstract

As Large Language Models (LLMs) evolve from passive text generators to active reasoning agents capable of interacting with external tools, the Model Context Protocol (MCP) has emerged as a key standardized framework for dynamic tool discovery and orchestration. Despite its widespread industry adoption, existing evaluation methods do not adequately assess tool utilization capabilities under this new paradigm. To address this gap, this paper introduces MCP-Radar, the first comprehensive benchmark specifically designed to evaluate LLM performance within the MCP framework. MCP-Radar features a challenging dataset of 507 tasks spanning six domains: mathematical reasoning, web search, email, calendar, file management, and terminal operations. It quantifies performance based on two primary criteria: answer correctness and operational accuracy. To closely emulate real-world usage, our evaluation employs both authentic MCP tools and high-fidelity simulations of official tools. Unlike traditional benchmarks that rely on subjective human evaluation or binary success metrics, MCP-Radar adopts objective, quantifiable measurements across multiple task domains, including computational resource efficiency and the number of successful tool-invocation rounds. Our evaluation of leading closed-source and open-source LLMs reveals distinct capability profiles and highlights a significant trade-off between accuracy and efficiency. Our findings provide actionable insights for both LLM developers and tool creators, establishing a standardized methodology applicable to the broader LLM agent ecosystem. All implementations, configurations, and datasets are publicly available at `https://anonymous.4open.science/r/MCPRadar-B143`.

## 1 Introduction

The paradigm of Large Language Models (LLMs) is undergoing a fundamental shift, evolving from passive text generators into proactive reasoning agents capable of interacting with external tools and APIs (Chowdhery et al., 2022; Brown et al., 2020). This evolution has been significantly accelerated by the advent of the Model Context Protocol (MCP), which provides a standardized framework for dynamic tool discovery and orchestration (Int; Mod; Qwe). As MCP adoption becomes widespread, the development of rigorous, standardized benchmarks to evaluate model performance within this new paradigm is critically important.

However, existing evaluation methodologies are insufficient. While traditional benchmarks excel at assessing knowledge-based reasoning (Hendrycks et al.) (Zhong et al.) or instruction following (Wang et al.) (Shridhar et al.), they offer limited insight into tool-use capabilities. Current tool-centric evaluations suffer from two primary limitations: 1) they struggle to differentiate between a model's genuine problem-solving via tools and mere recitation of pre-trained knowledge, 2) their reliance on simulated environments often fails to capture the complexities of real-world tool interactions.

To address these gaps, we introduce MCP-Radar, the first comprehensive benchmark designed specifically to evaluate LLM performance in the MCP paradigm. As illustrated in Figure 1, our methodology is structured around three core stages. First, in the Data Construction phase, we curate a diverse MCP Pool using both real-world tools from platforms like Smithery and high-fidelity mock MCPs for common applications such as email and calendar management. This pool supports two

distinct task categories: Precise Answer tasks (e.g., Math, Websearch), which have a single correct ground-truth value, and Fuzzy Match tasks (e.g., Filemanagement, Terminal), which require a correct sequence of operations.

Next, in the Test & Evaluation stage, we evaluate ten leading open- & closed-source LLMs. Our novel framework moves beyond simple binary success metrics by assessing accuracy through two core methods: Answer Matching for Precise Answer tasks and Operation Matching for Fuzzy Match tasks. Based on this, we quantify performance across multiple dimensions: Answer Accuracy (**RA**), Tool Selection Efficiency (**DTSR**), and Computational Resource Efficiency (**CRE**).

Our evaluation using MCP-RADAR reveals critical insights. For instance, while closed-source models significantly outperform open-source counterparts in mathematical reasoning, this gap narrows to less than 10% in web search tasks. More importantly, we identify a recurring failure pattern: models frequently select a semantically plausible but functionally incorrect tool, indicating a superficial understanding of the task requirements. Based on these findings, we provide actionable recommendations for both LLM development and MCP tool design.

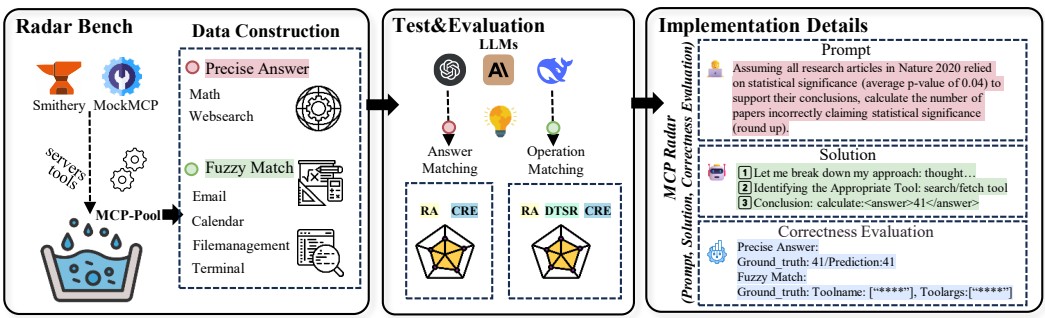

Figure 1: Overview of MCP-RADAR.

Our main contributions are threefold:

- We introduce MCP-RADAR, a comprehensive benchmark featuring two fundamental task types (Precise Answer and Fuzzy Match) across six critical domains: Mathematical Reasoning, Web Search, Email, Calendar, File Operations, and Terminal.
- We establish a high-fidelity evaluation environment by employing a combination of real-world MCP tools and meticulously replicated mock tools based on official specifications.
- We propose a novel, multi-dimensional evaluation framework for tool-augmented LLMs that utilizes purely objective and quantifiable metrics to assess accuracy, efficiency, and resourcefulness.

## 2 RELATED WORK

### 2.1 THE EVOLUTION TOWARDS STANDARDIZED TOOL USE

The Model Context Protocol (MCP) for Large Language Models (LLMs) is a unified interaction standard proposed by Anthropic to address systematic challenges in tool invocation. Early LLMs such as the GPT family relied only on static training data and were unable to access real-time information or interact with external systems, resulting in limited applications. Wei et al. (2022) demonstrated the role of structured reasoning in improving the performance of LLMs. With the increase of complex scenarios such as multi-round dialog systems, developers try to connect to external via API (Liu et al., 2024) (Song et al., 2023) (Qin et al., 2023) (Tang et al., 2023) to external data sources. Use the tool-enhanced LLM (Patil et al., 2024) (Parisi et al., 2022) (Lu et al., 2023) to try to solve the web browsing (Schick and Schütze, 2020) (Spiegel and Horák) (Chowdhury and Chowdhury, 2024) or code interpretation (Zhuang et al., 2023) (Liu et al., 2023) and other aspects of relevance, but Schick et al. (Schick et al., 2023) points out that this "peer-to-peer" integration leads to $N \times M$ issues, limiting system expansion and increasing maintenance costs. While platforms such as Hugging Face

promote model sharing, and frameworks such as LangChain attempt to enhance model capabilities through Tool Calling, these solutions do not address the underlying problem. However, these solutions still fail to address the fundamental problem. Hsieh et al. (2023) noting that these approaches still lack a common context delivery mechanism. It is in this context that the Model Context Protocol (MCP) was formally introduced and open sourced.

## 2.2 EVALUATING TOOL AND MCP PROFICIENCY

Evaluating the tool-use capabilities of LLMs has emerged as a critical research direction. While traditional evaluation frameworks focused on language comprehension, the advent of tool-augmented AI (Wang et al., 2023; Schick et al., 2023) has made specialized benchmarks for tool proficiency essential (Xu et al., 2023; Liang et al., 2024; Patil et al., 2024). The HELM framework, proposed by Liang et al. (2022), sought to establish multi-dimensional evaluation criteria but did not specifically address the efficiency and effectiveness of tool interaction protocols. Existing tool-use benchmarks exhibit several limitations: some struggle to handle complex scenarios such as long-context memory, multi-turn, or multi-tool calls (Li et al., 2023; Patil et al., 2024; Xu et al., 2023; Zhuang et al., 2023; Tang et al., 2023; Qin et al., 2023), while others rely on single-path, standardized answers that do not align with the diversity of real-world user needs (Wang et al., 2024). Furthermore, the dataset in Luo et al. (2025) is limited to operational-matching tasks, and the one in Liu et al. (2025) is entirely synthetic, raising doubts about its real-world applicability. Consequently, these systems lack a comprehensive and systematic evaluation of a model's ability to utilize specific protocols like MCP. In this paper, we introduce MCP-RADAR, a large-scale instruction benchmark, to explore the performance of LLMs in a variety of real-world MCP usage scenarios.

## 3 MCP-RADAR DATA GENERATION

The MCP-RADAR benchmark is comprised of 507 instances meticulously crafted to span six distinct real-world domains. To comprehensively evaluate the diverse capabilities of LLM agents, we structured our dataset around two fundamental task archetypes: Precise Answer and Fuzzy Match. A detailed breakdown of the instance distribution and the specific tools associated with each domain is provided in Table 1.

The two task categories are defined as follows: **Precise Answer:** This category includes tasks that require the model to return a single, definitive ground-truth value, such as a number, an algebraic expression, or a specific noun. As detailed in Table 1, this category covers the Math and Websearch domains. To ensure robustness and relevance, the instances for these tasks were curated and adapted from established academic datasets. Each data point consists of a query and its unique, verifiable answer. **Fuzzy Match:** This category encompasses operational tasks where success is determined not by a simple textual response, but by the correct invocation of an external tool with the appropriate parameters. This is essential for evaluating an agent's ability to act upon instructions in domains like Email, Calendar, Filemanagement, and Terminal. For these tasks, each data point consists of a query paired with the ground-truth tool name and its corresponding arguments. The step-by-step methodology for generating these goal-oriented instances is illustrated in Figure 2, with concrete examples available in subsection A.1.

| Task Type | Data-Domain | Quantity | #Tools | Tools |
|---|---|---|---|---|
| Precise Answer | Math | 120 | 4 | Calculate, SolveEquation, Differentiate... |
| | Websearch | 94 | 2 | Search, FetchContent |
| Fuzzy Match | Email | 119 | 17 | SendEmail, DraftEmail, ReadEmail... |
| | Calendar | 28 | 4 | ListCalendars, ListEvents, CreateEvent... |
| | Filemanagement | 91 | 13 | ReadTextFile, ReadMediaFile, ReadMultipleFiles... |
| | Terminal | 63 | 9 | GetConfig, SetConfigValue, StartProcess... |

Table 1: Data-Tool Statistics.

## 3.1 PRECISE ANSWER DATA CURATION

Our methodology for the Precise Answer dataset prioritizes answer accuracy, real-world relevance, and challenging queries. To achieve this, we chose to adapt and repurpose existing authoritative

datasets for mathematics and web search, rather than generating synthetic data from scratch (Zhuang et al., 2023) or deriving tasks solely from tool definitions (Styles et al., 2024). This approach grounds our benchmark in previously validated problems.

The curation process involved several key steps:

1. Data Sourcing and Filtering: We began by selecting the most challenging queries from high-quality source datasets (Gou et al., 2024; Fan et al., 2024; Srivastava et al., 2023; Kazemi et al., 2025; Mialon et al.). To ensure our benchmark specifically tests tool-use rather than a model's internal knowledge—a common issue of data contamination—we used a powerful baseline model (Gemini 2.5 Flash (Google)) as a filter. Queries that the model could solve without external tools were discarded, isolating problems that genuinely necessitate tool invocation.

2. Ground-Truth Annotation: For the remaining queries, we manually annotated the scope of potentially applicable tools for each problem. This annotation defines a set of valid tools without enforcing a single, rigid solution path. Crucially, for this task category, our evaluation focuses solely on the correctness of the final standard answer; the specific tools used are not assessed, only the result.

3. Tool Implementation: To execute these tasks, we integrated verified, open-source MCP tools. Specifically, we utilized the calculator-mcp-server for mathematical operations and the duckduckgo-mcp-server for web search functionalities.

## 3.2 FUZZY MATCH DATA CURATION

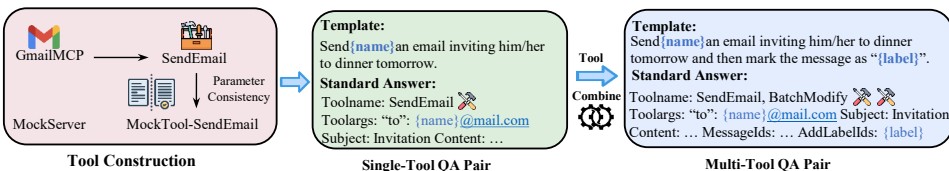

Figure 2: Data Generation.

The generation of the Fuzzy Match dataset involved two core stages: establishing a controlled tool environment and programmatically generating single- and multi-tool interaction scenarios.

**Tool Implementation and Environment**   To ensure reproducible and monitorable experiments, we developed a high-fidelity, semi-sandboxed tool environment. For the Filemanagement and Terminal domains, we integrated robust, community-developed open-source MCP tools. For the Email and Calendar domains, we implemented our own mock tools, EmailMCP and CalendarMCP, which meticulously replicate the interfaces and parameter structures of their real-world counterparts (GmailMCP and macOS CalendarMCP, respectively). These mock tools interact with a controlled, local database, pre-populated with 100 email and 50 calendar entries, rather than executing live operations. This setup provides realistic tool interaction schemas while maintaining a controlled evaluation environment. The email data format is detailed in subsection A.1.

**Instance Generation Methodology**   Our approach employs a template-based programmatic method, similar to frameworks like Workbench (Styles et al., 2024) and ToolQA (Zhuang et al., 2023), to generate question-and-action pairs.

For single-tool instances, we designed a unique template for each tool, from which five distinct tasks were generated. This process co-generates both the user query and the corresponding ground-truth tool invocation (toolname and toolargs), which forms the basis for our Fuzzy Match evaluation. Examples of these templates are provided in subsection A.1.

For multi-tool instances, we adopted a more constrained approach to avoid the combinatorial explosion of exhaustive tool pairings. We identified the top three most frequently used tools within each domain and created chained-task templatessubsection A.1 by combining their respective single-tool templates.

To ensure a unique and verifiable solution, the sequence of tool invocations in these multi-tool scenarios is strictly defined in the ground truth. The distribution of multi-tool questions per domain is shown in Figure 3.

# 4 EXPERIMENT

## 4.1 EXPERIMENTAL SETUP

**Models Evaluated** Our evaluation encompasses a diverse suite of ten leading Large Language Models, accessed via the OpenRouter API for standardized interfacing. The selection includes six state-of-the-art closed-source models: openai/gpt-5 (OpenAI), openai/gpt-4o (Hel), google/gemini-2.5-flash (Google), google/gemini-2.5-pro (Google), anthropic/claude-sonnet-4 (Anthropic), and anthropic/claude-3.7-sonnet (Cla); and four prominent open-source models: qwen/qwen3-235b-a22b-2507 (Alibaba Cloud, 2025), deepseek/deepseek-chat-v3-0324 (Yang), deepseek/deepseek-r1-0528 (DeepSeek AI), and meta-llama/llama-4-maverick (Meta AI, 2025).

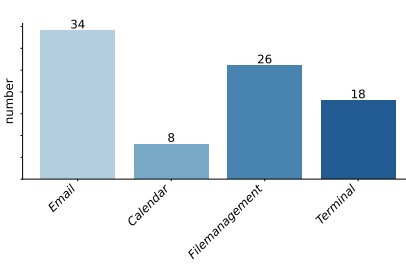

Figure 3: Multi-Tool Data Distribution.

**Implementation Details** Each model was tasked with solving problems using a set of 49 MCP tools distributed across the six domains. For every task, the model was provided with a system_message, the user question, and a list of available tools. The models were required to complete each task within a maximum of K=10 interaction rounds; exceeding this limit was considered a task failure.

The required output format depended on the task type. For Fuzzy Match tasks, the evaluation focused on the sequence of tool invocations generated by the model. For Precise Answer tasks, models were instructed to enclose their final response within a designated tag: <answer>[YOUR FINAL ANSWER]</answer>, ensuring unambiguous extraction of the answer. To mitigate tool-related hallucinations (Huang et al., 2025) and improve reliability, the system_message included detailed, tool-specific instructions. The complete prompt templates, along with a comparative analysis of different prompting strategies (e.g., ReAct vs. concise), are available in subsection A.2.

## 4.2 EVALUATION METRICS

Our evaluation framework employs distinct sets of metrics tailored to the unique success criteria of each task category.

**Precise Answer Tasks** For this category, evaluation focuses exclusively on the final outcome, as the tool-use path to a correct answer is often non-unique and may involve self-correction from erroneous steps. Consequently, we do not assess the intermediate tool invocation sequence. Performance is measured along two dimensions:

- **Result Accuracy (RA):** A binary metric indicating whether the model's final, extracted answer matches the ground truth exactly.
- **Computational Resource Efficiency (CRE):** A measure of the computational cost (e.g., token consumption) incurred to reach the solution.

We only require that the tools used by the model are from the valid set provided for the task, but we do not penalize alternative or redundant tool paths as long as the final answer is correct.

**Fuzzy Match Tasks** For tasks where the goal is to perform a correct operation, we evaluate the tool invocation process itself. Performance is assessed across three dimensions:

- **Result Accuracy (RA):** A binary metric indicating whether the model's final, decisive tool invocation (both tool name and parameters) exactly matches the ground-truth operation.

Table 2: Comparison of model performance metrics across two domains of MCP-RADAR. Gemini-Flash and Gemini-Pro are based on Gemini 2.5 Flash and Gemini 2.5 Pro respectively. Scores highlighted in red indicate the lowest score, while scores in green are the highest.

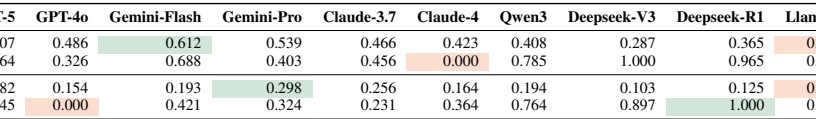

| Task | | GPT-5 | GPT-4o | Gemini-Flash | Gemini-Pro | Claude-3.7 | Claude-4 | Qwen3 | Deepseek-V3 | Deepseek-R1 | Llama-4 |
|------|------|-------|--------|--------------|------------|------------|----------|-------|-------------|-------------|---------|
| Math | ACC. | 0.607 | 0.486 | 0.612 | 0.539 | 0.466 | 0.423 | 0.408 | 0.287 | 0.365 | 0.128 |
| | CRE. | 0.564 | 0.326 | 0.688 | 0.403 | 0.456 | 0.000 | 0.785 | 1.000 | 0.965 | 0.644 |
| Websearch | ACC. | 0.182 | 0.154 | 0.193 | 0.298 | 0.256 | 0.164 | 0.194 | 0.103 | 0.125 | 0.008 |
| | CRE. | 0.245 | 0.000 | 0.421 | 0.324 | 0.231 | 0.364 | 0.764 | 0.897 | 1.000 | 0.965 |

Table 3: Comparison of model performance metrics across Four domains of MCP-RADAR

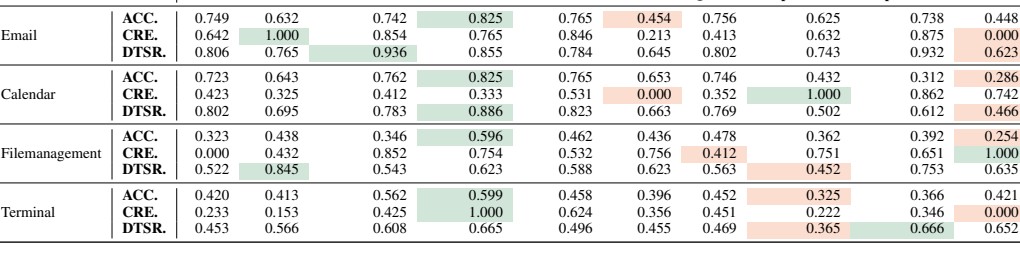

| Task | | GPT-5 | GPT-4o | Gemini-Flash | Gemini-Pro | Claude-3.7 | Claude-4 | Qwen3 | Deepseek-V3 | Deepseek-R1 | Llama-4 |
|------|------|-------|--------|--------------|------------|------------|----------|-------|-------------|-------------|---------|
| Email | ACC. | 0.749 | 0.632 | 0.742 | 0.825 | 0.765 | 0.454 | 0.756 | 0.625 | 0.738 | 0.448 |
| | CRE. | 0.642 | 1.000 | 0.854 | 0.765 | 0.846 | 0.213 | 0.413 | 0.632 | 0.875 | 0.000 |
| | DTSR. | 0.806 | 0.765 | 0.936 | 0.855 | 0.784 | 0.645 | 0.802 | 0.743 | 0.932 | 0.623 |
| Calendar | ACC. | 0.723 | 0.643 | 0.762 | 0.825 | 0.765 | 0.653 | 0.746 | 0.432 | 0.312 | 0.286 |
| | CRE. | 0.423 | 0.325 | 0.412 | 0.333 | 0.531 | 0.000 | 0.352 | 1.000 | 0.862 | 0.742 |
| | DTSR. | 0.802 | 0.695 | 0.783 | 0.886 | 0.823 | 0.663 | 0.769 | 0.502 | 0.612 | 0.466 |
| Filemanagement | ACC. | 0.323 | 0.438 | 0.346 | 0.596 | 0.462 | 0.436 | 0.478 | 0.362 | 0.392 | 0.254 |
| | CRE. | 0.000 | 0.432 | 0.852 | 0.754 | 0.532 | 0.756 | 0.412 | 0.751 | 0.651 | 1.000 |
| | DTSR. | 0.522 | 0.845 | 0.543 | 0.623 | 0.588 | 0.623 | 0.563 | 0.452 | 0.753 | 0.635 |
| Terminal | ACC. | 0.420 | 0.413 | 0.562 | 0.599 | 0.458 | 0.396 | 0.452 | 0.325 | 0.366 | 0.421 |
| | CRE. | 0.233 | 0.153 | 0.425 | 1.000 | 0.624 | 0.356 | 0.451 | 0.222 | 0.346 | 0.000 |
| | DTSR. | 0.453 | 0.566 | 0.608 | 0.665 | 0.496 | 0.455 | 0.469 | 0.365 | 0.666 | 0.652 |

- **Dialogue Turn Success Rate (DTSR):** Defined as the ratio of successful tool invocations to the total number of interaction turns. A "successful invocation" is one where the model selects an applicable tool and provides correctly formatted parameters, measuring its step-by-step ability to extract information and construct valid calls.

- **Computational Resource Efficiency (CRE):** A measure of the computational cost, with values normalized using max-min scaling to allow for cross-model comparison.

## 5 RESULTS

### 5.1 MAIN RESULTS

**Precise Answer Tasks** As shown in Table 2, closed-source models generally exhibit superior performance in this category. The performance gap is most pronounced in mathematical reasoning. The Websearch domain proved to be highly challenging for all models, with success rates universally below 30%. This difficulty stems from the dual requirement of selecting the correct tool and formulating a precise query to extract the necessary information. Among the models tested, Gemini-2.5-Pro emerged as the top performer with an accuracy of 29.8%, whereas the open-source Llama-4 recorded the lowest at 0.8%. While closed-source models maintained an advantage in Websearch, the performance gap narrowed compared to other tasks, with an average success rate of 20.7% versus 10.8% for open-source models.

**Fuzzy Match Tasks** In this category, model performance strongly correlates with task complexity. Models achieved significantly higher accuracy on simpler operational tasks (Email, Calendar) compared to more complex domains requiring precise sequential logic (Filemanagement, Terminal). A critical finding, detailed in Table 3, is the significant disparity observed between Dialogue Turn Success Rate (DTSR) and final accuracy (ACC) in complex tasks. For instance, in the Filemanagement domain, GPT-4o achieved a high DTSR of 84.5% but an ACC of only 43%. This 40.7% gap highlights a crucial failure mode: models can syntactically execute tool calls correctly but fail to select the semantically appropriate tool to solve the problem. This suggests a superficial understanding of the task's core requirements.

To further probe the models' planning capabilities, we conducted a targeted multi-tool experiment inspired by Huang et al. (2023). We tested whether providing a hint about the number of required tools would improve performance. The results in Table 4 show that such prompts had minimal impact, yielding only a 2.5% to 5% improvement. This indicates that the primary capability boundary for

Table 4: Multi Tool Selection Result. 2/2 means that we suggest call two tools, and then the LLM call the correct two tools. 2/ means that we do not advise about tools, and the LLM called the correct two tools. 1/2 means that on the basis of suggestion, the LLM called two tools, but only one of them was correct. 1/1 means that, based on the suggestion, the LLM only call one tool and it is correct

| Task | | GPT-5 | GPT-4o | Gemini-Flash | Gemini-Pro | Claude-3.7 | Claude-4 | Qwen3 | Deepseek-V3 | Deepseek-R1 | Llama-4 |
|---|---|---|---|---|---|---|---|---|---|---|---|
| | 2/ | 0.465 | 0.418 | 0.500 | 0.511 | 0.523 | 0.441 | 0.465 | 0.383 | 0.372 | 0.232 |
| ACC | 2/2 | 0.511 | 0.430 | 0.651 | 0.662 | 0.534 | 0.430 | 0.500 | 0.383 | 0.441 | 0.255 |
| | 1/1 | 0.023 | 0.047 | 0.023 | 0.000 | 0.023 | 0.047 | 0.058 | 0.023 | 0.058 | 0.070 |
| | 1/2 | 0.279 | 0.302 | 0.186 | 0.232 | 0.186 | 0.000 | 0.256 | 0.349 | 0.360 | 0.349 |

current models is not determining if a tool is needed, but rather deciding which specific tool to invoke and how to parameterize it correctly.

**Overall Performance** The holistic view presented in the radar charts Figure 4 reveals a distinct trade-off between performance and efficiency across the model landscape. While leading closed-source models demonstrate robust and well-rounded capabilities, certain open-source models achieve competitive accuracy, often at the cost of significantly higher computational resource (token) consumption. Notably, Gemini-2.5-Pro stands out as a highly capable tool-user across diverse domains. Among open-source models, Qwen demonstrates a commendable balance between task accuracy and resource efficiency.

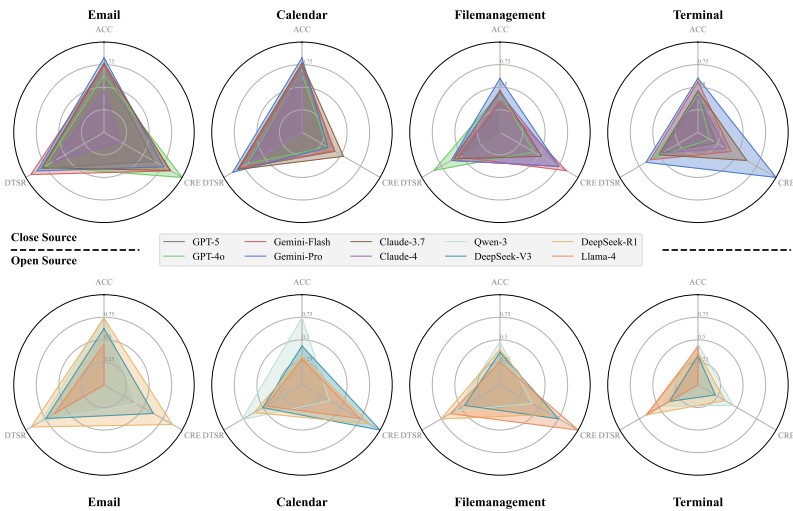

Figure 4: Model Performance Comparison Across Tasks. Longer edges indicate superior performance in each metric.

## 5.2 ABLATION STUDY

In our main experiments, we impose a limit on the maximum number of interaction rounds ($K$) to balance task performance with computational efficiency. However, this constraint could potentially limit a model's capacity for complex reasoning, reflection, and self-correction. To investigate this trade-off and justify our choice of $K$, we conducted an analysis on a randomly selected 50% subset of our dataset.

The results, illustrated in Figure 5, demonstrate a clear trend. As the maximum number of allowed rounds $K$ increases, the overall task accuracy for most models improves. This is expected, as more rounds allow for more attempts and corrective actions. However, we observe a point of diminishing returns. For most models, the rate of accuracy improvement slows considerably when $K \geq 10$, eventually beginning to plateau.

Therefore, considering the balance between maximizing solution accuracy and maintaining reasonable interaction latency, we selected $K = 10$ as the standard setting for all our main experiments.

# 6 ANALYSIS AND DISCUSSION

## 6.1 ERROR ANALYSIS

Our analysis identifies three primary categories of failures: Tool-Use Errors, Reasoning Errors, and Information Synthesis Errors.

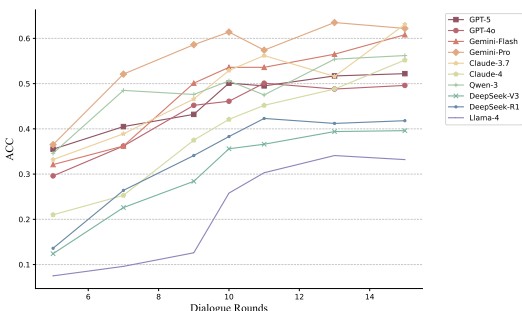

Figure 5: Impact of Dialogue Rounds on Average Accuracy Across Domains.

***Tool-Use Errors*** This category concerns failures in the direct invocation and selection of tools.**Parameter Error.** Occurs when the model selects the correct tool but supplies improperly formatted or invalid arguments. Examples include providing an invalid email address format or a malformed mathematical expression to a calculator (see Appendix B). **Inaccurate Tool Invocation.** Occurs when the model correctly identifies the need for a tool but selects one that is inappropriate for the given task. This often stems from a misinterpretation of a tool's functionality or its operational scope, such as using a basic arithmetic calculator for a problem requiring symbolic differentiation.

***Reasoning Errors*** This category includes failures in the model's high-level planning and logical deduction.**Tool Omission.** The model incorrectly assesses its own capabilities and attempts to solve a problem from its parametric knowledge when it should have invoked an external tool. This is common in complex tasks where the model fails to decompose the problem into tool-solvable sub-problems (see Appendix B). **Faulty Reasoning.** The model generates illogical or factually incorrect conclusions, even when the underlying tool outputs are accurate. A typical case is when a tool returns a correct number, but the model's final answer violates the problem's logical constraints (e.g., providing a decimal for a quantity that must be an integer), indicating a failure to integrate tool outputs with the problem's semantic context. **Redundant Tool Invocation.** The model becomes trapped in a reasoning loop, repeatedly invoking the same or similar tools without making substantive progressAppendix B. This behavior suggests deficiencies in its planning and state-tracking capabilities, as it fails to update its strategy based on new observations.

***Information Synthesis Errors*** This category involves failures in processing and utilizing the information returned by tools. **Tool-Result Integration Error.** The model obtains a correct intermediate result from a tool but fails to integrate it into subsequent reasoning steps. For instance, a model might correctly solve an equation with a calculator but then fail to substitute the result back into a larger derivation. **Information Extraction Failure.** The model successfully retrieves a large volume of information (e.g., from a web search) but fails to extract, filter, or summarize the core information relevant to the query. This manifests as either presenting irrelevant content or providing a disorganized data dump instead of a synthesized answer.

***Other Types*** this type of error accounts for a relatively small proportion. It mainly includes errors such as interaction termination due to excessive interaction rounds, incorrect result solutions, or the invocation of non-existent tools.

For tasks requiring precise answers, such as complex web searches or mathematical problems, Faulty Reasoning and Tool Omission are the most prevalent failures. In the case of niche web queries, models often struggle to grasp the key points, leading to a reliance on their internal knowledge base which can result in factual hallucinations. For mathematical tasks, models often misjudge their own capability boundaries and, viewing the reasoning as overly cumbersome, will attempt to solve problems without invoking the necessary tools.

Conversely, for tasks involving fuzzy matching or complex tool parameterization, direct execution errors are more frequent. In these cases, Parameter Errors are the most common issue, particularly

when a tool requires a large number of arguments, increasing the likelihood of incorrect data entry (e.g., placing a recipient's email in the subject line). Furthermore, persistent instances of Inaccurate Tool Invocation in these scenarios suggest that LLMs retain fundamental misunderstandings about the specific application scope of certain tools.

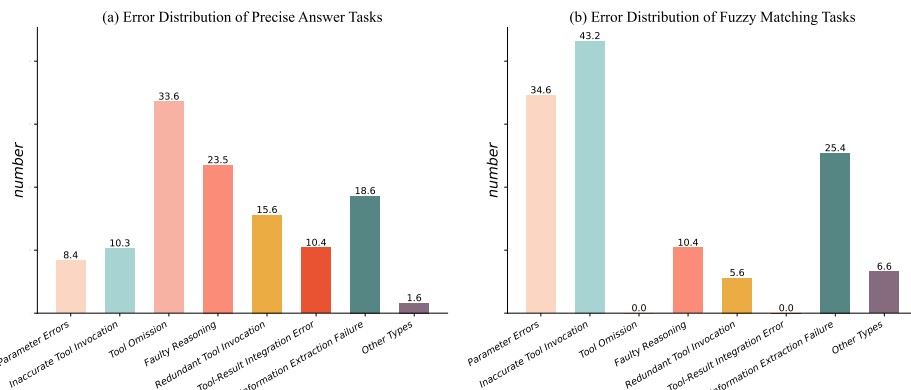

Figure 6: Error Distribution by Task Type.

## 6.2 DISCUSSION AND CONCLUSION

In this paper, we introduced MCP-RADAR, a comprehensive framework to systematically evaluate the tool-use capabilities of Large Language Models within the MCP paradigm. By assessing ten leading models across six domains using a combination of real-world and high-fidelity mock tools.

Our findings reveal a critical gap between a model's syntactic ability to invoke a tool and the deeper semantic understanding required to solve problems effectively. We consistently observed that while models can often execute a tool call, they struggle with the higher-level reasoning required for proactive and precise tool selection, multi-step planning, and effective information synthesis from tool outputs. These core challenges point to clear directions for both model and tool development.

**Implications for LLM Development**

- **Improving Proactive Tool Invocation:** Current models exhibit a tendency to default to their parametric knowledge, failing to recognize their own capability boundaries. Future training should focus on improving this self-awareness, encouraging models to view external tools as a primary resource rather than a last resort.

- **Fostering De-compositional Reasoning:** We observed that models often attempt to solve complex problems with a single tool call, particularly in domains like advanced mathematics.Enhancing their ability to deconstruct tasks into a sequence of smaller, tool-solvable steps is crucial for tackling multi-stage problems.

**Recommendations for MCP Tool Developers**

- **Optimizing Tool Descriptions:** A tool's description is a critical interface for the LLM. Descriptions must be both concise and precise, as overly verbose or ambiguous text significantly increases the model's cognitive load and leads to invocation errors.

- **Promoting Atomic Tool Design:** Our results suggest that LLMs are more proficient at orchestrating a sequence of simple, single-purpose ("atomic") tools than understanding and correctly parameterizing a complex, multi-functional one. Developers should favor creating granular tools that can be combined to solve complex tasks.

## ETHICS STATEMENT

The authors confirm their adherence to the ICLR Code of Ethics. This research introduces MCP-RADAR, a benchmark for evaluating the tool-use capabilities of Large Language Models (LLMs), and does not propose a new model architecture. Our primary goal is to foster transparency and guide the responsible development of AI agents by providing the community with objective, quantifiable evaluation metrics.

We acknowledge that advancing the capabilities of tool-using agents carries an inherent dual-use risk; more competent agents could potentially be repurposed for malicious activities. Our work aims to mitigate such risks by providing a clear framework for identifying model weaknesses, such as the observed tendency for models to select incorrect tools, which can inform the development of safer and more reliable systems.

The datasets used in MCP-RADAR are constructed from established public benchmarks (e.g., MATH, GAIA) or generated programmatically. The underlying LLMs evaluated may reflect societal biases present in their training data. While our benchmark measures performance, it does not explicitly address or mitigate these biases, which remains a critical area for future research. To prevent real-world harm during evaluation, operational tasks involving tools like email, calendar, and terminal commands were conducted in a semi-sandboxed environment, using custom-built mock servers that replicate tool functionality without executing real operations. We believe the benefit of a standardized, objective evaluation framework for agentic models significantly contributes to the safe and ethical progression of AI.

## REPRODUCIBILITY STATEMENT

We have taken extensive measures to ensure the reproducibility of our work. The complete implementation of the MCP-RADAR benchmark, including all configurations, evaluation scripts, and the full dataset, has been made publicly available at https://anonymous.4open.science/r/MCPRadar-B143.

The data generation process is described in detail in Section 3. This includes the repurposing of existing public datasets for "Precise Answer" tasks (Section 3.1.1) and the template-based programmatic generation for "Fuzzy Match" tasks (Section 3.1.2). Appendix A provides further examples of data templates and prompts. The specific models evaluated are listed in Section 4.1, and the novel, quantifiable evaluation metrics (Result Accuracy, Dialogue Turn Success Rate, and Computational Resource Efficiency) are formally defined in Section 4.2.

Our experimental setup, including system prompts and the maximum interaction rounds, is detailed in Section 4.1. The tools used in the benchmark are a combination of open-source MCP servers and custom-built mock tools designed to replicate official specifications, with sources and implementation details provided in https://anonymous.4open.science/r/MCPRadar-B143. We believe these resources provide a comprehensive basis for replicating our results and extending this research.

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

## A DATASET AND EXPERIMENT DETAILS

### A.1 DATASET DETAILS

For precise answer type query:

```
{"unique_id": "db4fd70a-2d37-40ea-873f-9433dc5e301f",
 "Prompt": "As of May 2023, how many stops are between South
    Station and Windsor Gardens on MBTAs Franklin-Foxboro
    line (not included)?",
 "Answer": "10"}
```

For fuzzy match type query:

```
{"unique_id": "db4fd70a-2d37-40ea-873f-9433dc5e301f",
 "Prompt": "Please send Lucy@gamil.com an email for me,
    invite her to come over for dinner tomorrow.",
 "Toolname": "SendEmail",
 "Toolargs": "from":(default)mock@localhost.com, "to":
    Lucy@gamil.com, "subject": ".", "body": "I hope this
    message finds you well. I would like to warmly invite you
    to come over for dinner at my place tomorrow evening."}
```

Emaildata Format:

```
"email_id":"c98db9b0-347b-4357-b7fd-0075f944acb3","status":"outbox",
"to":["Harmon@mail.com"],
"subject":"hihihihihi","body":"nice to meet you",
"cc":[],"bcc":[],"attachments":[],
"sent_datetime":"2025-08-19T09:43:46.979Z","labels":["important"]
```

Single-Tool Template Format:

```
Email:
"Please email {name} to notify her/him to attend the meeting."
Calendar:
"Please add {something} to the schedule."
Filemanage:
"Please return the content in {path} file."
Terminal:
"Start {python scripts}"
```

Multi-Tool Template Format Example:

```
Email:
"Please email {name} to notify her/him to attend the meeting

and then mark the message as {label}"
```

### A.2 SYSTEM PROMPT

In this section, we demonstrate the impact of setting the prompt to concise mode versus ReAct format on experimental results.

**ReAct Version:**

Table 5: Prompt Effect Comparison

| Mean-Acc. | GPT-5 | GPT-4o | Gemini-Flash | Gemini-Pro | Claude-3.7 | Claude-4 | Qwen3 | Deepseek-V3 | Deepseek-R1 | Llama-4 |
|---|---|---|---|---|---|---|---|---|---|---|
| ReAct | 0.524 | 0.386 | 0.499 | 0.512 | 0.533 | 0.462 | 0.602 | 0.388 | 0.374 | 0.282 |
| Concise | 0.501 | 0.461 | 0.536 | 0.614 | 0.529 | 0.421 | 0.506 | 0.356 | 0.383 | 0.258 |
| Δ | -0.023 | 0.075 | 0.037 | 0.102 | -0.004 | -0.041 | -0.096 | -0.032 | 0.009 | -0.024 |

You are a professional mathematics assistant that must solve problems by following a loop of Thought -> Action -> Observation. Your sole tool is calculator-mcp-server.
**Tools** You can only use calculator-mcp-server.
calculator-mcp-server calculate: Evaluates a mathematical expression and returns the result. solve_equation: Solves algebraic equations for x and returns all solutions. differentiate: Computes derivatives of expressions. integrate: Computes indefinite integrals of expressions.
**Instructions** Strictly follow the ReAct loop: Thought -> Action -> Observation until the problem is solved.
Every action must be a call to the calculator-mcp-server tool.
The action format must be an exact JSON string.
The final answer must be enclosed in <answer> tags.
**Format** Thought: You must describe your reasoning process in detail here.
Analyze the problem and determine what needs to be solved.
Plan the steps to solve the problem.
Decide which tool function to call and what its input parameters should be.
Action: Call the tool. You must use the following JSON format.
JSON
"server_name": "calculator-mcp-server", "tool_name": "...", "inputs": ...    Observation: The output result from the tool.
**Example** Question: What is the determinant of matrix [[4, 7], [2, 6]]?
**Thought:** The question asks for the determinant of a matrix. I must use the calculator-mcp-server tool to perform this calculation. I will use the calculate function, and its expression parameter will be det([[4, 7], [2, 6]]).
**Action:**
JSON
"server_name": "calculator-mcp-server", "tool_name": "calculate",
"inputs":  "expression": "det([[4, 7], [2, 6]])"

**Concise Prompt:**

"You are a specialized mathematics assistant. " "You MUST ONLY use the calculator-mcp-server's tools to solve math problems to solve all math problems. " "Do NOT provide any direct answers without using tools. " "Always analyze the problem first and then call calculator-mcp-server's tools with the corresponding arguments. " "Show your reasoning steps clearly and format answers using the tool's output. " "You MUST format your final answer as: <answer>[YOUR FINAL ANSWER]</answer>" "Mathematical answers should be numbers or formatted in LaTeX, e.g.,

$$\begin{pmatrix} 31 & 50 \\ -18 & -29 \end{pmatrix}$$

We randomly selected ten datasets from each domain and tested them with two different prompts. The results are shown in Table 5.

## B  RELATED EXPERIMENT RESULTS

Error Type:

- Tool Omission

User: "Suppose $f(x) = x^2$, and $g(x)$ is a polynomial such that $f(g(x)) = 4x^2 + 4x + 1$. Enter all possible polynomials $g(x)$, separated by commas."

Assistant: "We are given that: - $f(x) = x^2$ - $g(x)$ is a polynomial - $f(g(x)) = (g(x))^2 = 4x^2 + 4x + 1$ So we need to find all polynomials $g(x)$ such that:

$$(g(x))^2 = 4x^2 + 4x + 1$$

### Step 1: Analyze the right-hand side Note that:

$$4x^2 + 4x + 1 = (2x + 1)^2$$

So,

$$(g(x))^2 = (2x + 1)^2$$

Taking square roots (keeping in mind that polynomials can be positive or negative), we get:

$$g(x) = 2x + 1$$

or

$$g(x) = -(2x + 1) = -2x - 1$$

These are both polynomials, and squaring either gives:

$$(2x + 1)^2 = 4x^2 + 4x + 1$$

### Step 2: Are there any other polynomial solutions? Suppose $g(x)$ is a polynomial such that $(g(x))^2 = (2x + 1)^2$. Then:

$$(g(x))^2 - (2x + 1)^2 = 0 -> (g(x) - (2x + 1))(g(x) + (2x + 1)) = 0$$

So the polynomial equation implies that $g(x) - (2x + 1) = 0$ or $g(x) + (2x + 1) = 0$, since a polynomial cannot be zero at all points unless one of the factors is identically zero (as polynomials form an integral domain). Thus, the **only** polynomial solutions are:

$$g(x) = 2x + 1 quad text or quad g(x) = -2x - 1$$

### Final Answer: <answer>2x + 1, -2x - 1</answer>"

- Redundant Tool Invocation. It repeatedly queries the same website address

User: I'm researching species that became invasive after people who kept them as pets released them. There's a certain species of fish that was popularized as a pet by being the main character of the movie Finding Nemo. According to the USGS, where was this fish found as a nonnative species, before the year 2020? I need the answer formatted as the five-digit zip codes of the places the species was found, separated by commas if there is more than one place.

Assistant: The species of fish popularized by the movie *Finding Nemo* is the clown anemonefish (*Amphiprion ocellaris*). "tool_use_detail": [["type": "call", "content": "[Calling tool search with args 'query': 'Finding Nemo fish invasive species USGS', 'max_results': 5]", "type": "result", "content": "tool return result meta=None content=[TextContent(type='text', text=Found 5 search results:1. Nonindigenous Aquatic Species URL: https://nas.er.usgs.gov/ ***URL: https://www.usgs.gov/index.php/centers/ wetland-and-aquatic-research-center/science/science-topics/ nonindigenous-aquatic-species Summary: TheUSGSNonindigenous AquaticSpecies(NAS) database tracks occurrence data on non-native aquatic plant and animalspeciesthroughout the United States, and provides the public with-speciesprofiles, distribution maps, and online/real-time queries for state/hydrologic basin-specificspecies. 3. Invasive Species Experts Database | U.S. Fish & Wildlife Service ****URL: https://www.usgs.gov/index.php/centers/wetland-and-aquatic- research-center/science/science-topics/nonindigenous-aquatic-species

- Parameter Errors

```
User: Please send an email to Jimmy inviting him to dinner tonight.
Assistant: "email_id":"c98db9b0-347b-4357-b7fd-0075f944acb3",
"status":"outbox",
"to":[],
"subject":"Jimmy@mail.com",
"body":""Hey, want to come over for dinner tonight?","cc":[],"bcc":[],
"attachments":[],
"sent_datetime":"2025-09-19T09:43:46.979Z", "labels":[""]
```

Table 6: Impact of Dialogue Rounds K on Average Accuracy Across Domains)

| Dialogue Rounds (K) | GPT-5 | GPT-4o | Gemini-Flash | Gemini-Pro | Claude-3.7 | Claude-4 | Qwen3 | Deepseek-V3 | Deepseek-R1 | Llama-4 |
|---|---|---|---|---|---|---|---|---|---|---|
| K=5 | 0.355 | 0.296 | 0.321 | 0.365 | 0.332 | 0.210 | 0.346 | 0.124 | 0.136 | 0.075 |
| K=7 | 0.405 | 0.362 | 0.362 | 0.521 | 0.389 | 0.253 | 0.485 | 0.226 | 0.264 | 0.096 |
| K=9 | 0.432 | 0.452 | 0.501 | 0.586 | 0.466 | 0.375 | 0.476 | 0.284 | 0.341 | 0.126 |
| K=10 | 0.501 | 0.461 | 0.536 | 0.614 | 0.529 | 0.421 | 0.506 | 0.356 | 0.383 | 0.258 |
| K=11 | 0.495 | 0.501 | 0.536 | 0.574 | 0.562 | 0.452 | 0.475 | 0.366 | 0.423 | 0.303 |
| K=13 | 0.517 | 0.488 | 0.565 | 0.635 | 0.516 | 0.488 | 0.554 | 0.394 | 0.412 | 0.341 |
| K=15 | 0.522 | 0.496 | 0.608 | 0.622 | 0.631 | 0.552 | 0.562 | 0.396 | 0.418 | 0.332 |

## C  EXPERIMENTAL PROCEDURE FORMATTING

```
MCP-Pool
"mcp_pool": [
{
"name": "FireCrawl",
"description": "A Model Context Protocol (MCP) server implementation that integrates with Firecrawl for
web scraping capabilities.",
"tools": [
{
"tool_name": "firecrawl_search",
"tool_description": "Search the web and optionally extract content from search results.",
"inputs": [
{
"name": "query",
"type": "string",
"required": true,
"description": "your search query"
}
]
}
],
"run_config": [
{
"command": "npx -y firecrawl-mcp",
"env": {
"FIRECRAWL_API_KEY": "your key"
},
"port": your port
}
]
}
```

**Eval-Result**

```
{
    "unique_id": "*****",
    "question": "Given a integer n(>0), make a pile of n levels of ...",
    "ground_truth": "def make_a_pile(n):\n
return [n + 2*i for i in range(n)]",
    "prediction": "def make_a_pile(n):\n
pile = []\n for i in range(n):\n
if n % 2 == 0:\n pile.append(n + 2*i)\n
else:\n pile.append(n + 2*i)\n
return pile",
    "success": true,
    "tool_usage": {
      "tool_calls": [
        {"name": "read_file", "arguments": "{\"path\":\problem.jsonl\"}"},
        {"name": "write_file", "arguments": "{\"path\":\answer.jsonl\"}"}
      ],
      "total_tool_count": 2,
      "tool_names": ["read_file", "write_file"]
    },
    "token_usage": {
      "prompt_tokens": 820,
      "completion_tokens": 610,
      "total_tokens": 1430
    }
}
```

## D  THE USE OF LARGE LANGUAGE MODELS

Large Language Models (LLMs) were utilized in two capacities during this research: dataset construction and manuscript preparation.

First, as detailed in Section 3.1, LLMs played an auxiliary role in the creation of the MCP-RADAR benchmark. For the precise answer tasks, we utilized results from Gemini 2.5 Flash to help select queries from existing datasets that required tool invocation. For the fuzzy match tasks, question-answer pairs were programmatically generated based on author-designed templates, a process inspired by LLM-based data generation methodologies.

Second, Gemini 2.5 Pro was employed as a writing assistant to polish the manuscript by improving grammar, refining phrasing, and enhancing overall clarity.

All core scientific contributions, including the research ideation, the design of the evaluation framework and metrics, experimental setup, and final analysis, were performed exclusively by the human authors. The authors have carefully reviewed all machine-generated content and take full responsibility for the validity, integrity, and originality of this entire work.

