# OpenReview forum: "MCP-Radar: A Multi-Dimensional Benchmark for Evaluating Tool Use Capabilities in Large Language Models"
_ICLR.cc/2026/Conference — Submitted to ICLR 2026_

### Official Review · Reviewer_yHUx · 2025-10-30

**Soundness:** 3
**Presentation:** 3
**Contribution:** 1
**Rating:** 4
**Confidence:** 3

**Summary:**

This paper presents MCP-Radar, a benchmark for evaluating the tool-use capabilities of Large Language Models within the Model Context Protocol (MCP) framework. It features a diverse set of 507 tasks across six domains, categorized into "Precise Answer" and "Fuzzy Match" objectives. The benchmark employs a combination of real-world and high-fidelity mock tools to create a realistic testing environment. The authors propose a multi-dimensional evaluation framework assessing accuracy, efficiency, and step-by-step success. A key finding is that models often struggle with semantically correct tool selection despite being syntactically proficient at invocation, providing useful insights for future development.

**Strengths:**

1. The paper reveals critical, actionable issues in how LLMs interact with tools, such as the tendency to select a "semantically plausible but functionally incorrect tool." This raises important questions for the community, for instance, whether tool metadata is sufficient for LLMs to interpret correctly, especially since models cannot pre-train on the functionality of every possible MCP. The analysis provides useful, concrete insights for improving MCP design, such as recommending atomic tool design and optimized descriptions.

2. The construction of the benchmark is intuitive. The use of both real and mock tools creates a high-fidelity yet controlled evaluation setting. The distinction between "Precise Answer" and "Fuzzy Match" tasks effectively captures different facets of tool-use capability. The data curation process, which filters out problems solvable without tools, successfully isolates and tests genuine tool-use necessity.

3. The writing is generally clear and well-structured. The methodology and results easy to follow.

**Weaknesses:**

1. The paper's claim to be the "first comprehensive benchmark" for evaluating LLMs in the MCP paradigm is overstated, as several precedent works exist. For example, "MCP-AgentBench" (Guo et al., arXiv:2509.09734) and the concurrently developed "MCP-Universe" (Luo et al., 2025) and "MCPEval" (Liu et al., 2025) presented in the paper's own related work section, directly tackle similar problems of MCP-based agent evaluation. While MCP-Radar is authentic and useful, its incremental novelty over these recent benchmarks may not be sufficient to stand out in the rapidly growing landscape of agent evaluation works.

2. Though the benchmark is well-executed and will be valuable for model training and evaluation, the core concept of a new tool-use benchmark, even within the specific MCP framework, no longer feels novel or exciting enough for a top-tier conference to me, given the booming quantity of such works. I am left uncertain whether this specific contribution meets the bar for acceptance.


**Minor Issues**

1. Many citations are formatted inconsistently. For instance, line 39 uses "APIs (Chowdhery et al., 2022; Brown et al., 2020)" while line 45 uses "knowledge-based reasoning (Hendrycks et al.) (Zhong et al.)". There are also formatting errors, such as in line 103: "API (Liu et al., 2024) (Song et al., 2023)..." should use a single set of parentheses.

2. Figure 4 (Model Performance Comparison) is very unclear and difficult to interpret in its current form.

3. Typo: Line 421: "Other Types this type"

**Questions:**

1. Given the existence of "MCP-Universe," "MCPEval," and "MCP-AgentBench," how does MCP-Radar provide a fundamentally unique or more comprehensive evaluation that justifies its "first" claim and distinguishes it significantly from these precedents?

2. The finding that models struggle with semantic tool selection is crucial. Did your analysis offer any specific, data-backed recommendations for what constitutes a "good" vs. "bad" tool description that could help MCP developers mitigate this issue?

3. The ablation study on interaction rounds (K) is helpful. Were any experiments conducted concerning the impact of different tool description styles or detail levels on model performance, especially in light of the observed "Inaccurate Tool Invocation" errors?

---

> ### Author Response · Authors · 2025-11-21
>
> We thank you for your review and insightful feedback. We have carefully considered each point and provide our responses below.
>
> **1. Novelty**
>
> We acknowledge the rapid pace of development in this field and the existence of concurrent works. We have revised the manuscript to position MCP-RADAR not just as "first," but as the first multi-dimensional diagnostic benchmark specifically anchored in High-Fidelity Mock Environments. Our fundamental distinctiveness lies in two key areas:
>
> - High-Fidelity Mock Environments: This is our most significant differentiator. Most existing benchmarks (including the ones cited) rely largely on static QA pairs or stateless API calls where the "result" is just text. In contrast, we developed MockMCP servers (e.g., EmailMCP, CalendarMCP) backed by local databases. This allows us to perform Operation Matching, i.e. verifying actual state changes (e.g., "Did the email actually appear in the outbox table with the correct metadata?") rather than just parsing the model's text output. This offers a level of ecological validity that purely text-based benchmarks cannot match.
>
> - Multi-Dimensional Metrics: We go beyond accuracy by introducing a dual-track system (Precise Answer vs. Fuzzy Match) combined with Computational Resource Efficiency (CRE). This allows us to benchmark the "economic efficiency" (tokens per success) of agents, a dimension often absent in the cited works.
>
> **2. Practical Recommendations for Tool Design**
>
> Our error analysis (Section 5) provides actionable recommendations for MCP developers, derived directly from observed failure modes:
> - We observed a high frequency of Inaccurate Tool Invocation errors (43.2%) in tasks involving complex, multi-functional tools. Our data suggests that models struggle to parameterize "Swiss Army Knife" style tools. Developers should favor granular, single-purpose ("atomic") tools. Models are significantly more proficient at chaining a sequence of simple tools (e.g., `search_files` $\rightarrow$ `read_file`) than correctly configuring one highly complex tool with many optional parameters.
>
> - Our qualitative analysis of semantic failures indicates that overly verbose or ambiguous descriptions increase the model's cognitive load, leading to hallucinations. Tool descriptions should be concise and precise, focusing strictly on the direct action and required arguments, avoiding unnecessary conversational filler or vague capability claims.
>
> **3. Tool description styles**
>
> This is an excellent suggestion. While our initial submission included an ablation on Prompting Strategy (ReAct vs. Concise, Table 5), we agree that an ablation on Description Style is valuable. Based on your feedback, we have committed to adding the following experiment in the next version:
>
> We will select a subset of challenging Fuzzy Match tasks and evaluate model performance under three conditions: (1) Standard Description (Baseline), (2) Minimalist/Atomic Description, and (3) Verbose Description. This will provide quantitative evidence to support our recommendation that description optimization is a key lever for mitigating semantic selection failures.
>
> **4. Minor Issues and Formatting**
>
> We have rigorously standardized all citation styles throughout the manuscript (e.g., merging multiple citations into single parentheses). We have redesigned Figure 4 to use a non-overlapping, stratified chart type to significantly enhance interpretability and visual clarity. The typo in Line 421 ("Other Types this type") and other minor errors have been corrected.

---

### Official Review · Reviewer_X6ka · 2025-11-01

**Soundness:** 2
**Presentation:** 3
**Contribution:** 2
**Rating:** 4
**Confidence:** 4

**Summary:**

This paper proposes a benchmark to evaluate LLMs’ tool-use capability within the MCP. The benchmark covers 507 tasks across six domains and introduces a set of metrics. The authors evaluate 10 closed-source and open-source LLMs and provide an analysis of tool-use errors, reasoning errors, and efficiency trade-offs.

**Strengths:**

1. This paper evaluates the tool-use capability of LLM Agents with MCP.
2. The detailed analysis of errors (tool-use errors, reasoning errors, information synthesis errors) provides useful insights into current LLM Agent limitations.

**Weaknesses:**

Despite being positioned as an MCP-focused benchmark, MCP-RADAR's overall structure largely resembles that of prior tool-use evaluation benchmarks.

**Questions:**

1. What is the core novelty of MCP-RADAR compared to other benchmarks?
2. Real MCP usage involves tools being added, removed, or replaced dynamically. Can MCP-RADAR evaluate LLMs under dynamic or expanding tool sets?

---

> ### Author Response · Authors · 2025-11-21
>
> We thank you for your review and insightful feedback. We have carefully considered each point and provide our responses below.
>
> **1. Novelty**
>
> While MCP-RADAR builds upon the foundational logic of general tool-use evaluations, it introduces three specific innovations designed to address the limitations of current benchmarks (e.g., ToolBench, API-Bank):
>
> - Unlike prior benchmarks that rely on ad-hoc Python functions or generic REST API definitions, MCP-RADAR is the first benchmark grounded in the Model Context Protocol (MCP). This distinction is critical because MCP is rapidly emerging as the industry standard for connecting LLMs to data (supported by Anthropic, Replit, and others). Our benchmark specifically evaluates a model's ability to adhere to this standardized interoperability layer, which generic function-calling benchmarks do not capture.
>
> - Many existing benchmarks rely on static QA pairs or stateless API calls. In contrast, we developed High-Fidelity MockMCP servers (e.g., EmailMCP, CalendarMCP). These are stateful environments backed by local databases that mimic real-world counterparts. This allows us to perform operation matching, i.e. verifying not just the text output, but the actual state change in the system (e.g., "Did the email actually land in the outbox with the correct label?"). This offers a significantly higher level of ecological validity than purely text-based evaluations.
>
> - We introduce a dual-track evaluation system distinguishing between Precise Answer (result-oriented) and Fuzzy Match (process-oriented) tasks. Furthermore, by integrating our Computational Resource Efficiency metric, we provide a holistic view that balances performance accuracy against the economic cost of the agent.
>
> **2. Evaluation of Dynamic or Expanded Toolsets**
>
> This is an excellent point. While real-world MCP usage involves dynamic discovery (e.g., `list_tools` handshakes), our current experimental setup utilized a standardized search space to ensure fair, reproducible comparisons across models. However, the architecture of MCP-RADAR is inherently designed to support dynamic complexity.
>
> In our experiments, models are not given a single isolated tool; they must select the correct tool from a diverse pool of 49 valid MCP tools loaded into the context. This effectively simulates the post-discovery phase, where an agent must distinguish between relevant tools and irrelevant ones .
>
> Because our benchmark is built on actual, modular MCP server implementations (rather than static text descriptions), the framework natively supports dynamic tool injection. The MCP-Pool architecture allows researchers to dynamically add, remove, or swap servers in the agent's environment during runtime to test adaptability.
>
> We agree that explicitly testing the discovery phase where the tool set changes mid-conversation is a valuable direction. We will add a discussion in the final version emphasizing that MCP-RADAR’s modular server architecture serves as a ready-to-use testbed for such dynamic evaluation scenarios.

---

> ### Comment · Reviewer_X6ka · 2025-11-26
>
> Thanks to the authors for the response, which has addressed my concerns.

---

### Official Review · Reviewer_Br9z · 2025-11-01

**Soundness:** 2
**Presentation:** 2
**Contribution:** 2
**Rating:** 2
**Confidence:** 5

**Summary:**

MCP-RADAR is a benchmark for evaluating LLMs’ tool-use within the Model Context Protocol, introducing 507 tasks across six domains. It measures both answer correctness and operational accuracy using real MCP tools and high-fidelity simulations, with objective metrics like resource efficiency and successful invocation rounds. Evaluations of leading open- and closed-source LLMs reveal distinct capability profiles and an accuracy–efficiency trade-off, and the artifacts are publicly available.

The main contribution of this paper is a comprehensive MCP-RADAR benchmark with two task types across six domains, a high-fidelity evaluation testbed using real and spec-accurate mock MCP tools, and an objective multi-metric framework measuring accuracy, efficiency, and resourcefulness.

**Strengths:**

The strengths of the paper can be summarized as,

- The paper’s code and data are open-sourced, which benefits community progress in this area and helps ensure the work’s credibility and reproducibility.

- Proposes a benchmark spanning six domains and two task types.

**Weaknesses:**

The weaknesses of the paper are listed as follow,

- Introduction Section:

    - Causal overclaim about MCP. The authors say the shift to tool-using agents was “significantly accelerated by the advent of the Model Context Protocol (MCP)” and that MCP is “a standardized framework for dynamic tool discovery and orchestration.” That’s disputable: tool use predates MCP (e.g., function calling, ReAct/Toolformer, agent frameworks), and MCP is one proposal among several—not an established, consensus “standard,” nor the main accelerator.

    - Overgeneralization about tool-centric evals. The authors claim they “rely on simulated environments,” but there are evaluations using real websites/APIs/browsers. The point (simulations miss real-world friction) is fair, but stated too absolutely.

    - “First comprehensive benchmark” is an overclaim. There are many tool-use/agent benchmarks already. Even if it is MCP-specific, calling it “first” and “comprehensive” needs careful qualification or evidence.

    - Web search ≠ single ground truth. The authors place “Websearch” under Precise Answer with a single correct value, but the web is non-static, multi-answer, and time-dependent—this undermines reproducibility and the stated task type.

    - Operation-matching may mis-score valid solutions. For “Fuzzy Match”, the authors require “a correct sequence of operations.” In many tools, multiple sequences are functionally equivalent (commutativity, idempotent ops, retries). Step-matching will falsely mark correct runs wrong unless the authors define state-equivalence.

    - Cross-model comparability of CRE is dubious. Closed-source APIs and local open-source models have very different latency/tokenization/rate-limit/hardware stacks. Without strict budget control and normalization, Computational Resource Efficiency comparisons are not apples-to-apples.

    - Real + mock tools confound conclusions. Mixing Smithery “real-world” MCPs with “high-fidelity mock MCPs” in one score can entangle environment quality with model performance. Results may reflect simulator fidelity, not the model.

    - Naming inconsistency. The authors alternate between “MCP-RADAR” and “Radar Bench” in the figure 1—could confuse readers about what is the benchmark vs the implementation component.

- Data Generation Section:

    - Arithmetic inconsistency. The authors state “507 instances,” but Table 1’s quantities sum to 515 (120+94+119+28+91+63). That undermines all downstream stats.

    - Self-contradictory definition of “Precise Answer.” It’s defined as a “single, definitive ground-truth value,” yet examples include algebraic expressions and SolveEquation—both admit many equivalent forms (and sometimes multiple solutions).

    - “Comprehensive” claim vs narrow coverage. Six domains with ~500 items is useful but not “comprehensive” for MCP agents.

    - Selection-by-one-model bias + non-reproducibility. Gating items with Gemini 2.5 Flash (“discard queries the model could solve without tools”) bakes one closed model’s strengths/weaknesses into the dataset and cannot be replicated or audited by others.

    - Construct invalidity: claims to test tool-use but scores tool-agnostic. The authors say the goal is to “specifically test tool-use,” yet for Precise Answer the authors only score the final answer and “do not assess the specific tools used.” A stronger model can bypass tools entirely—so the data do not measure the construct the authors claim.

    - Unverifiable, drifting web environment. Tying execution to a “duckduckgo-mcp-server” without specifying snapshotting, locale, time, rate-limit, or version pinning makes results non-deterministic and unreproducible across sites and dates.

    - Goal/tool mapping from source datasets is under-specified. The authors say items are “adapted from established datasets,” then executed via MCP tools. The pipeline that maps each source problem to which tool(s) and why (and how equivalence classes are formed) is not described—this is a core methodological gap.

    - Tiny, unrealistic world state. Pre-populating 100 emails / 50 calendar entries is far from real usage and encourages pattern learning (names, subjects, label sets), weakening claims of “realistic tool interaction.”

    - A “unique template per tool” with five variants may measure template recognition, not tool reasoning.

    - Arbitrary “top-3 tools” selection. “Most frequently used” is asserted but not grounded (logs? prior studies?). With mocks and synthetic tasks, “frequency” becomes circular and may bias chains toward your chosen endpoints.

- Experiments Section:

    - DTSR represents Tool Selection Efficiency in introduction section and denotes Dialogue Turn Success Rate in Experiments Section, which is confusing.

    - Ill-posed error taxonomy (overlap/ambiguity). “Tool-Use,” “Reasoning,” and “Information Synthesis” aren’t mutually exclusive (e.g., “tool omission,” “redundant tool invocation,” and “tool-result integration” can co-occur). Without an operational coding scheme and inter-rater agreement, counts are unreliable.

    - Figure 5 invites causal misinterpretation. “Impact of Dialogue Rounds on Accuracy” suggests causality, but there’s strong selection confounding (easy tasks finish in fewer rounds; harder ones allow/need more rounds; models differ in round budgets). No stratification, controls, or CIs are shown.

**Questions:**

Please refer to the Weaknesses box for details. In summary, the paper’s motivation for the dataset, its definition, construction, and experimental analysis all contain substantial flaws. The main claimed novelty—Fuzzy Match—reads more like a variant of binary matching and does not clearly distinguish its evaluation metrics from those of similar datasets. The data appear to be simply gathered from existing sources and reprocessed, so the workload is questionable. Moreover, possible bias relative to real-world distributions further weakens the dataset’s contribution.

---

> ### Author Response · Authors · 2025-11-21
>
> We thank you for your review and insightful feedback. We have carefully considered each point and provide our responses below.
>
> **1. Overclaim**
>
> We have revised the manuscript to describe MCP as a "promising and rapidly emerging framework" that facilitates tool orchestration, rather than claiming it is the sole accelerator or an established global standard.
>
> We have refined our claims to state that MCP-RADAR is the "first benchmark specifically tailored to the MCP ecosystem," rather than a general "first" for all tool-use agents. We have clarified the text to acknowledge that while real-world evaluations exist, they often face significant challenges in reproducibility (due to dynamic web states) and safety. MCP-RADAR aims to bridge these gaps through its high-fidelity sandboxed design.
>
> **2.Precise answer definition**
>
> We appreciate this insight regarding the dynamic nature of the web. However, our methodology mitigates this risk. As detailed in Section 3.1 (Precise Answer Data Curation), our curation strategy deliberately avoids dynamic queries (e.g., weather, flight prices). Instead, we strictly selected questions targeting objective, static facts (e.g., "What is the color of the heroine's dress in the movie Big Hero 6?"). This approach aligns with the methodology of established benchmarks like GAIA, ensuring that the ground truth remains stable over time regardless of when the web search is performed.
>
> **3. Fuzzy match**
>
> We acknowledge that multiple paths can exist, but our design controls for this. To ensure objective evaluation, we designed the prompts to be constraint-based. As described in Section 3.1, our multi-tool prompts explicitly guide the logical flow (e.g., "First mark the calendar for the important event, and then email Lisa..."). These constraints ensure that the ground-truth sequence is the only logically valid path for the given instruction, minimizing the risk of penalizing alternative valid solutions.
>
> **4. Cross-model comparability of CRE**
>
> We believe our CRE comparison is valid and standardized. We utilize OpenRouter as a unified gateway to standardize the interface for both open-source and closed-source models. By tracking token consumption through this unified layer, we measure the economic and computational efficiency (tokens per task) rather than hardware-dependent metrics like latency or wattage. This provides a fair comparison of the reasoning cost across different architectures, independent of the underlying serving infrastructure.
>
> **5. Real + mock tools**
>
> We clarify that our "Mock" tools are effectively High-Fidelity Sandboxed Implementations. Testing email/calendar agents on live production systems poses significant safety risks (e.g., accidental spamming or data corruption). Our simulated tools utilize the exact schemas and logic of real-world MCP servers (like GmailMCP) but operate on a local, controlled database. This allows us to measure the model's ability to use the tool correctly without ethical risks, ensuring both safety and high fidelity. The "simulation" is only in the execution backend, not the API interface the model interacts with.
>
>
> **6. Inconsistency**
>
> We have unified the terminology to "MCP-RADAR" throughout the paper and figures in the revised manuscript.
>
> The correct total is 515 instances. We have corrected the typo in Table 1 and updated all derived statistics in the revision.
>
> **7. Self-contradictory definition of Precise Answer**
>
> While the definition emphasizes a definitive answer, our evaluation pipeline accommodates equivalent forms. We employ a combination of precise string matching, regex parsing, and LLM-as-a-Judge verification to ensure fairness for answers that are semantically identical but syntactically different (e.g., time formats or algebraic equivalents).
>
> **8. Model bias**
>
> Our filtering methodology follows standard practices in recent literature (e.g., Toolformer, GAIA). Using a strong model to filter "easy" samples ensures the benchmark measures tool-use necessity rather than internal memorization. While this introduces a dependency on the filter model, it effectively raises the difficulty floor of the benchmark. It is a necessary step to ensure that the dataset targets the "knowledge boundary" where tools are actually required.

---

> > ### Author Response · Authors · 2025-11-21
> >
> > **9. Construct invalidity**
> >
> > Our benchmark design explicitly accounts for this through a rigorous "Necessity Filter." As detailed in Section 3.1, we utilized a powerful baseline model (Gemini 2.5 Flash) to filter the dataset. Any query that the baseline could solve without external tools (relying solely on parametric knowledge) was discarded. The remaining dataset consists exclusively of problems that genuinely necessitate tool invocation (e.g., performing complex calculations beyond a tokenizer's arithmetic ability or retrieving specific real-time data). Therefore, if a model achieves the correct answer on this filtered dataset, it is a strong indicator that it successfully utilized the tool, as the problem is effectively unsolvable otherwise.
> >
> > **10. Goal/tool mapping**
> >
> > The mapping was deterministic and intent-based, not arbitrary. During the Data Construction phase (Section 3), we first classified the intent of each source question (e.g., "Solving a quadratic equation" or "Searching for a specific PDF"). Based on this intent, we mapped the question to the specific tool in our MCP-Pool capable of satisfying that intent. This ensures a one-to-one alignment between the problem's semantic requirements and the tool's functional capabilities. We have added a subsection "Source-to-Tool Mapping Strategy" in the revision to make this transparent.
> >
> > **11. Tiny, unrealistic world state**
> >
> > We acknowledge the current database size is relatively small compared to enterprise scale. However, 100 items are sufficient to strictly test search logic, parameter retrieval, and filtering capabilities. We plan to expand the magnitude of the background data in future iterations to test scalability, but the current size effectively differentiates model performance on logic and reasoning.
> >
> > **12. Templates measure recognition, not reasoning**
> >
> > We acknowledge that templates introduce structural regularity, but they do not compromise the evaluation of Tool Reasoning for three reasons:
> >
> > - Models must distinguish between semantically similar tools within the same domain (e.g., `ReadTextFile` vs. `ReadMediaFile` vs. `ReadMultipleFiles`). A simple template trigger is insufficient; the model must reason about embedded contextual variables to select the precise tool.
> > - A successful invocation requires both correct selection and correct parameter extraction. Guessing the tool name via overfitting yields a score of 0 if the model fails to extract complex parameters from the non-templated variable parts.
> > - For Multi-Tool templates (e.g., "Check calendar THEN send email"), the model must perform decompositional reasoning to determine execution order and data flow, which simple pattern matching cannot resolve.
> >
> > **13. Top-3 tools**
> >
> > The Top 3 tools were selected based on popularity metrics from the official MCP registry and Smithery. This ensures we prioritize tools that agents are most likely to encounter in real-world deployments.
> >
> > **14. DTSR confusion**
> >
> > We have corrected the definition of DTSR (Dialogue Turn Success Rate) in the Experiments Section to align with the definition in Section 4.2.
> >
> > **15. Error taxonomy**
> >
> > We acknowledge the inherent complexity in categorizing errors where failures may cascade. To ensure reliability, we employed a hierarchical decision process during annotation:
> >
> > - Tool-Use Errors (Priority 1): Malformed API calls or wrong tool selection.
> > - Information Synthesis Errors (Priority 2): Correct data returned, but failure to extract/format it.
> > - Reasoning Errors (Priority 3): Perfect tool interaction, but flawed logical deduction based on results. In the final version, we will explicitly detail this scheme and include the Fleiss’ Kappa coefficient to quantitatively demonstrate inter-rater reliability.
> >
> > **16. Figure 5**
> >
> > We respectfully clarify that Figure 5 presents a controlled ablation study, not an observational correlation.
> > In this experiment, "Dialogue Rounds ($K$)" is not a dependent variable (i.e., we are not plotting how long a model chose to take). Rather, $K$ is an independent hyperparameter that we manipulated. We evaluated the same models on the same dataset subset while strictly enforcing different maximum round limits ($K=5, 7, \dots, 15$).
> > Since the dataset (and thus task difficulty) remained constant across all values of $K$, there are no confounding factors regarding task complexity. The observed improvement in accuracy is causally linked to the relaxation of the round constraint (budget), demonstrating that models perform better when afforded a larger budget for self-correction. We have revised the figure caption to explicitly state that $K$ represents the maximum allowed budget.
> >
> >
> > We hope these clarifications address your concerns and demonstrate the rigor and contribution of MCP-RADAR.

---

### Official Review · Reviewer_NX29 · 2025-11-05

**Soundness:** 3
**Presentation:** 2
**Contribution:** 3
**Rating:** 4
**Confidence:** 5

**Summary:**

This paper introduces MCP-RADAR, a comprehensive benchmark designed for the MCP and tool usage. The benchmark features 507 tasks across six domains. An evaluation of ten leading LLMs revealed a critical weakness: models often select semantically plausible but functionally incorrect tools, indicating a superficial understanding of the task.

**Strengths:**

MCP usage is a highly meaningful evaluation direction. Authors have constructed a diverse and challenging benchmark that even advanced models perform poorly on it. Moreover, their analysis of model behavior provides substantial value for future research.

**Weaknesses:**

- The authors did not compare against a baseline that does not use tool calling, making it unreasonable to claim a causal relationship between model performance and MCP tool usage.
- Some tasks have very few evaluation examples, like only 28 for calendar, which introduces instability in the evaluation. Repeated runs could yield significantly variances.
- Figure 4 and Table 3 are highly redundant and neither provides an average score to summarize each model’s overall performance across tasks.
- During evaluation, the authors included all tools in the context. An ablation study that restricts the context to only task-relevant tools would be necessary to assess the impact of tool scope on performance.
- The main body part contains too little useful information. Detailed data construction pipeline and data sources are lacked.

**Questions:**

- How the CRE metric calculated?
- The authors used tools such as web search. How robust are these tools? Could they return different results across different invocations, thereby affecting reproducibility?

---

> ### Author Response · Authors · 2025-11-21
>
> We thank you for your review and insightful feedback. We have carefully considered each point and provide our responses below.
>
> **1. Baseline**
>
> We respectfully clarify that a no-tool baseline was implicitly evaluated during our data construction phase. As detailed in Section 3.1 Precise Answer Data Curation, we employed a rigorous filter mechanism. We used a powerful baseline model (Gemini 2.5 Flash) to attempt all queries without tools. Any query that the model could solve using only parametric knowledge was discarded. Therefore, the final dataset consists exclusively of tasks where a no-tool baseline would theoretically achieve near 0% accuracy. We will make this zero-shot no-tool baseline implication more explicit in the experimental section to justify the causal link between MCP usage and performance.
>
> **2. Few evaluation examples**
>
> The limited size was due to the initial strict manual verification of the mock environment. In next version, we commit to expanding the dataset, particularly for the Calendar domain, using our programmatic generation pipeline to ensure statistical significance and reduce variance in repeated runs.
>
> **3. Redundant figure/table**
>
> We agree that the presentation can be streamlined. We will consolidate the redundant visualization and data tables. Crucially, as suggested, we will add a Macro Average score for each model across all domains in the final tables. This will provide a clear, single-metric summary of each model's overall tool-use capability, facilitating easier comparison.
>
> **4. Ablation study**
>
> We respectfully clarify that providing the full set of 49 tools was a deliberate design choice to ensure ecological validity. A core promise of the MCP framework is "dynamic tool discovery", where agents must autonomously identify relevant utilities from a crowded, realistic server environment. Restricting the context to only relevant tools would degrade the task to a simple slot-filling exercise, failing to capture the "Noise Filtering" capability that is critical for real-world MCP agents.
>
> However, we agree with the reviewer that discerning between Selection Failure (finding the tool) and Execution Failure (using the tool) is scientifically valuable. To address this, we will conduct an Ablation Study in the revised manuscript:
>
> - Method: We will re-evaluate a representative subset of models (e.g., GPT-4o, Claude-3.5, and Llama-4) in the Fuzzy Match domains.
> - Setup: In this control group, the System Prompt will explicitly contain only the ground-truth tools required for the specific task (Partial Context), removing all distractors.
> - Expected Insight: By comparing the performance gap between the Full Context (Main Result) and Partial Context (Ablation), we will quantify the Distraction Cost. This will allow us to pinpoint whether a model's limitation lies in retrieval or reasoning.
>
> **5. Data construction details**
>
> We appreciate this feedback regarding readability. We will move the critical details of the Data Construction Pipeline (currently in Appendix A) and the specific Source-to-Tool Mapping logic into the main body (Section 3). This will ensure the methodology is self-contained and reproducible without requiring constant reference to the appendix.
>
> **6. How the CRE metric calculated**
>
> Computational Resource Efficiency (CRE) is calculated based on the total token consumption required to solve a task. We utilized the standardized API provided by OpenRouter to track both input (prompt) and output (completion) tokens for every model interaction. We normalize these values using max-min scaling to ensure cross-model comparability. This metric serves as a proxy for the economic cost of reasoning.
>
> **7. Robustness**
>
> As described in Section 3.2, these are high-fidelity, sandboxed simulations. We have performed extensive testing to ensure they are deterministic and execution-safe, ensuring 100% reproducibility.
>
> While the web is dynamic, we mitigated variance by selecting queries targeting static facts rather than volatile real-time data. Our internal checks confirmed that the search tool consistently returns relevant hits for these queries, ensuring that performance fluctuations are due to model reasoning, not tool failure.

---

### Meta-Review · Area_Chair_mX7A · 2026-01-01

**Summary:**

The paper introduces MCP-Radar, a benchmark designed to evaluate LLM tool-use capabilities within the Model Context Protocol (MCP), but it is recommended for rejection due to reviewer consensus on its limited novelty relative to concurrent benchmarks (e.g., MCP-Universe), overstated claims regarding "comprehensive" coverage, and methodological issues with the Computational Resource Efficiency (CRE) metric.

It is also noted that the arxiv preprint version of this manuscript submission writes as published in ICLR'26, which imo is inappropriate and manipulative (it's under review rather than published).

**Reviewer Concerns:**

While the rebuttal successfully clarified the implicit "no-tool" baseline and the planned expansion of small domain datasets (e.g., Calendar), significant concerns remain outstanding regarding the fairness of comparing efficiency metrics across closed and open-source models and the paper's incremental contribution in a rapidly saturated field.

**Reviewer Scores:**

All reviewers consistently rated the paper as marginally below acceptance (Score: 4), and it is unlikely that further discussion would have resulted in a positive score shift given the fundamental and shared reservations about the work's distinguishability from existing tool-use benchmarks.

---

### Decision · Program_Chairs · 2026-01-26

Reject